# Exosomes—Promising Carriers for Regulatory Therapy in Oncology

**DOI:** 10.3390/cancers16050923

**Published:** 2024-02-25

**Authors:** Teresa Maria Jóźwicka, Patrycja Maria Erdmańska, Agnieszka Stachowicz-Karpińska, Magdalena Olkiewicz, Wojciech Jóźwicki

**Affiliations:** 1Department of Oncology, Faculty of Health Sciences, Ludwik Rydygier Collegium Medicum in Bydgoszcz, Nicolaus Copernicus University, 87-100 Torun, Poland; teresa.jozwicka@cm.umk.pl (T.M.J.); patrycja.e55@gmail.com (P.M.E.); 2Department of Lung Diseases, Tuberculosis and Sarcoidosis, Kuyavian-Pomeranian Pulmonology Center, 85-326 Bydgoszcz, Poland; askarpinska@gmail.com; 3Eurecat, Centre Tecnològic de Catalunya, Unitat de Tecnologia Química, Marcel·lí Domingo 2, 43007 Tarragona, Spain; magda.olkiewicz@eurecat.org; 4Department of Pathology, Kuyavian-Pomeranian Pulmonology Center, 85-326 Bydgoszcz, Poland

**Keywords:** cancer regulatory therapy, extracellular vesicles, exosomes, engineered exosomes, signaling molecules

## Abstract

**Simple Summary:**

Surgery, radiotherapy, and chemotherapy, the standard oncological treatments, are increasingly revealing their limitations. Population treatments are based on the percentage of patients that benefit, with no guarantee of sustained treatment effectiveness even when personalized and molecularly targeted therapies are considered. Cancer always develops with the “consent“ of the immune system, which is considered the guardian of the cancer process. Therapy aimed at tumor destruction does not recognize the subtleties associated with the likely coexistence of tumor tissue protection systems. A new concept of regulatory therapy, based on targeted interference in the mechanisms of molecular processes underlying carcinogenesis along with cancer promotion and progression, may represent a promising direction in the search for effective therapies. Recently, there has been increased interest in exosomes, which are the cellular products of intercellular communication and carriers of signaling molecules, based on the potential for regulatory interference in cancer process. We have knowledge about these mechanisms, but do we have the technology to allow such interference? If so, the time has come for regulatory therapy in oncology.

**Abstract:**

Extracellular vesicles (EVs), including exosomes and microvesicles, together with apoptotic bodies form a diverse group of nanoparticles that play a crucial role in intercellular communication, participate in numerous physiological and pathological processes. In the context of cancer, they can allow the transfer of bioactive molecules and genetic material between cancer cells and the surrounding stromal cells, thus promoting such processes as angiogenesis, metastasis, and immune evasion. In this article, we review recent advances in understanding how EVs, especially exosomes, influence tumor progression and modulation of the microenvironment. The key mechanisms include exosomes inducing the epithelial–mesenchymal transition, polarizing macrophages toward protumoral phenotypes, and suppressing antitumor immunity. The therapeutic potential of engineered exosomes is highlighted, including their loading with drugs, RNA therapeutics, or tumor antigens to alter the tumor microenvironment. Current techniques for their isolation, characterization, and engineering are discussed. Ongoing challenges include improving exosome loading efficiency, optimizing biodistribution, and enhancing selective cell targeting. Overall, exosomes present promising opportunities to understand tumorigenesis and develop more targeted diagnostic and therapeutic strategies by exploiting the natural intercellular communication networks in tumors. In the context of oncology, regulatory therapy provides the possibility of reproducing the original conditions that are unfavorable for the existence of the cancer process and may thus be a feasible alternative to population treatments. We also review current access to the technology enabling regulatory intervention in the cancer process using exosomes.

## 1. Introduction

The standard oncological treatments for the general population, namely surgery, radiotherapy, and systemic therapy, are increasingly revealing their limitations, even with the modifications of so-called personalized medicine. Cancer treatment is subject to limitations arising from clinical criteria such as performance status, organ function, toxicity, or patient preferences. Attempts to improve these, for example by combining with immunotherapy or applying molecular targeting, have not yet resulted in any therapeutic breakthroughs in oncology and therefore still pose a challenge [1]. The question arises: Does a therapeutic alternative to population-based treatment exist? There is increasing interest in the potential to acquire technologies enabling the controlled introduction of changes in the metabolism of cancer cells and/or their microenvironment, thus providing the basis for so-called regulatory cancer therapy. Exosomes can interact with recipient cells through surface receptor binding and the delivery of their bioactive cargo, thus influencing processes like proliferation, survival, and invasion [2]. In cancer, tumor-derived exosomes (TEXs) are key mediators between cancer cells and the tumor microenvironment. TEXs carry oncogenic proteins and nucleic acids that can promote tumor growth, metastasis, and therapy resistance [3]. For example, TEXs from aggressive cancer cells can induce the epithelial–mesenchymal transition (EMT), which is a process associated with increased metastasis [4]. TEXs can also polarize tumor-associated macrophages toward protumoral phenotypes that suppress anticancer immunity [5]. Overall, TEXs facilitate the remodeling of the tumor microenvironment to favor cancer progression. However, recent advances in exosome engineering methods present opportunities to develop TEX-based nanotherapeutics that counteract these pathological functions [6]. Exosomes are highly biocompatible and can bypass biological barriers by virtue of sharing surface proteins with host cells. Exogenous cargo, like chemotherapeutic drugs, RNA therapeutics, or tumor antigens, can be loaded into exosomes to achieve enhanced, targeted therapeutic delivery and antitumor immune activation [7]. In this review, recent insights into how TEXs influence the tumor microenvironment are summarized, thus highlighting the emerging strategies used in exosome modification to convert TEXs from being promoters into inhibitors of cancer progression. The ability to engineer anticancer TEXs could enable more precise and potent therapeutic interventions against aggressive cancers.

## 2. Exosomes—Variations and Biological Significance

Extracellular vesicles are microscopic, spherical biological structures released into the extracellular matrix from the membrane surface of most living cells [3]. All EVs are composed of a two-layer lipid membrane, within which are proteins, lipids, sugars, and nucleic acids (DNA or RNA) [8,9]. They are found present in body fluids such as fertile fluid, saliva, and urine, among others [10]. EVs are found in a wide range of sizes and differ in biogenesis and overall function. They mainly include exosomes, microvesicles (MVs), and apoptotic bodies [11,12]. Because of their ability to carry functional biological molecules and to integrate with target cells, EVs have become the subject of research over the past few decades regarding their potential use as a new therapeutic target [13,14,15]. The most widely studied group of EVs are exosomes. Exosomes are the smallest carriers belonging to EVs, with a diameter ranging from 30 to 150 nm [16]. They are produced in the extracellular matrix by many types of living cells, including dendritic cells, B cells, T cells, fat cells, epithelial cells, and cancer cells [8]. Exosomes are formed in multivesicular bodies (MVBs), which are then released from cell membranes via exocytosis. The process of their formation consists of several stages in which numerous proteins are involved, e.g., Rab GTPases, as well as endosomal sorting complexes required for transport—(ESCRT): ESCRT-0, ESCRT-1, ESCRT-2, and ESCRT-3, together with CD9, CD81, CD63, flotillin, TSG101, ceramide, and Alix—which are characteristic markers for exosomes [16,17]. In recent scientific reports, an alternative mechanism involved in exosomal biogenesis has been suggested that is independent of ESCRT and involves lipids, tetraspanins, and/or heat shock proteins [18]. A characteristic feature of exosomes is their heterogeneity. The population of these nanomolecules may vary from one another depending on their size, content, functional impact on recipient cells, and source of origin [17,19]. Exosomes are assigned a major role in cellular processes, including participating in angiogenesis, proliferation, cell aging, apoptosis, differentiation, and the transmission of immune signals [14,20]. Due to their ability to interact between proteins on the surface of exosomes and recipient cells, these molecules can be specifically used as carriers of information about the cancer microenvironment [2]. More and more findings from scientific research are indicating the potential for the use of exosomes as prognostic and predictive biomarkers of numerous cancers, e.g., in breast cancer and colon cancer [21,22], as drug carriers (pancreatic cancer and glioblastoma) [23,24], and as a form of targeted therapy—for example, as inhibitors that reduce the survival of cancer cells in pancreatic cancer [25].

## 3. Scientific Observations

### 3.1. Gene Expression and 5-Year Survival

Based on the analysis of data from the public HPA (Human Protein Atlas) database, we hypothesized that cancer malignancy, expressed in terms of 5-year survival, may depend on the expression of selected genes (Table 1). Gene expression levels in RNA-seq data are estimated using the FPKM (fragments per kilobase of script per million mapped reads) parameter, which is a measure that takes into account both gene length and the total number of mapped reads in a sample. Since the FPKM value is given for each gene, tumor, and case in the HPA database, we decided to check whether there is an expression value of selected genes that would allow for stratifying cases into two groups with a significantly different chance of 5-year survival. We assessed all cancer groups available in the database. The data presented in Table 1 show that an optimal FPKM value can be determined for selected genes, thereby allowing for the stratification of the patient population into groups with significantly different survival chances [26]. Indicating such an FPKM value as a survival threshold may be helpful in designing new technologies and in optimizing existing technologies for gene expression regulation. 

Therefore, if the prognosis of cancer may depend on the expression of one or even several proteins, this serves as justification to search for a way to modulate their expression and observe the co-occurrence of malignant changes in the cancer cell.

### 3.2. Remodeling of the Tumor Microenvironment 

Well-known elements of modeling the cancer microenvironment are the process of epithelial–mesenchymal transition of cancer cells [44] and the metabolic impact of cancer-associated fibroblasts (CAFs) and immune system cells, including tumor-associated macrophages (TAMs) [45]. 

#### EMT Regulators

Many signaling molecules can activate the so-called main regulators of the epithelial–mesenchymal transition (EMT). These include proteins from the Snail family (Snail1 and Snail2 (Slug)), Zeb proteins, and proteins from the TWIST family. Factors such as TGF-β, HIF1α, β-catenin, IL-6, caveolin-1, vimentin, and nucleic acids, including miRNAs, which are transported with the help of exosomes secreted by cancer cells, play an important role in the mobilization of EMT regulators [46]. Jing Cai et al. treated ovarian cancer cells with exosomes derived from the peritoneal fluid (ascites-derived exosomes: ADEs) of ovarian cancer patients and observed a morphological and immunohistochemical change from an epithelioid to a mesenchymal phenotype (elongated, spindle-shaped shape, decreased E-cadherin expression, increased N-cadherin, and vimentin expression) [38]. The analysis of RNA sequencing microarray data showed that an important element of the EMT transformation is miR-6780b-5p, which is transported with the help of ADEs and can also be promoted by exosomes from cancer stem cells (CSCs). The element miR-19b-3p transported in CSC exosomes from clear cell renal cell carcinoma (CCRCC) induced EMT by reducing PTEN protein expression. Exosomes from the CSCs of patients with lung metastases of clear cell carcinoma induced the strongest EMT-promoting effect. The percentage of CD103+ exosomes was higher in the tumor cells, peripheral blood, and lungs of these patients than those without lung metastases. Lu Wang et al. suggested that the CD103+ protein influences the “redirection” of CSC exosomes to cancer cells and organs, thus increasing the metastatic potential of CCRCC [40]. Another pathway leading to exosome-mediated epithelial–mesenchymal transformation in hepatocellular carcinoma (HCC), observed by Lu Chen et al., is the MAPK/ERK signaling pathway [47]. Cheng-Shuo Huang et al. identified two potential aggressiveness-promoting exosomal long noncoding RNAs: LINC00960 and LINC02470. Exosomes derived from high-grade bladder cancer cells increased the viability, migration, invasion, and clonogenicity of low-grade bladder cancer cells and activated major EMT signaling pathways, including the β-catenin, Notch, and Smad2/3 signaling pathways [48].

Observations of changes in the malignancy of the cancer process associated with changes in the expression of specific but distinct signaling proteins justify research related to identifying these factors, the so-called modulators, of their expression; some are well known, whereas others require definition.

A study conducted by Chen L. et al. showed that exosomes derived from liver cancer mediate the EMT process and increase the malignancy of cancer cells. The authors found that exosomes derived from the MHCC97H cancer line induced human lens epithelial cells (HLE cells) to the EMT process, as evidenced by the higher expression of mesenchymal markers α-SMA and vimentin and the lower expression of epithelial marker E-cadherin than in control HLE cells. Moreover, an increase in the expression level of EMT promoters (ZEB1, ZEB2, and Slug) and a decrease in the level of OVOL1, a promoter inducing mesenchymal–epithelial transition (MET), were observed in HLE cells treated with MHCC97H-derived exosomes. It was suggested that in liver cancer cells, the EMT process induced by exosomes may be mediated by the MAPK/ERK pathway, as evidenced by the increased phosphorylation of Erk1/2 proteins in HLE cells [49].

### 3.3. Cancer-Associated Fibroblasts (CAFs)

Tumor-associated fibroblasts are important for tumor progression, both functionally and structurally. There are three main sources of origin of CAFs: mesenchymal stem cells (MSCs), EMT cells, and cells residing in tissues. The source of CAFs in the tumor environment also appears to be pericytes, which, under the influence of extracellular vesicles produced by cancer cells, are probably transformed into cancer fibroblasts by inducing the PI3K/AKT and MEK/ERK pathway [50].

A cancer tumor needs metabolic support from the cellular elements of the microenvironment, which is provided by cancer-associated fibroblasts mobilized by the tumor through intercellular communication involving exosomes.

The promotion effect was examined in the experiments of Hu JL et al., which showed that CAFs directly promote colorectal cancer (CRC) cell metastasis and resistance to 5-FU/L-OHP drug therapy through secreted exosomes [29]. The process of penetration of exosomes derived from CAFs into cancer cells has been proven, in CRC cells, by an increase in the level of miR-92a-3p, whose role in the regulation of angiogenesis has already been confirmed in studies on gastric and esophageal cancer [50,51]. The importance of the involvement of CAF-derived exosomes in promoting CRC cell invasion and migration was confirmed in mouse models in the same in vivo study, where CRC cells were found to form lung metastases following treatment with a medium containing exosomes secreted by CAF [29].

The results of research conducted by C. R. Goulet et al. suggested that CAFs induce EMT-related changes in cancer cells mainly through the secretion of the cytokine IL-6. The exposure of RT4 bladder cancer cells to a culture medium with CAFs significantly induced the expression of N-cadherin, vimentin, SNAIL1, TWIST1, and ZEB1 while suppressing the expression of E-cadherin and phospho-ß-catenin. It was shown that CAFs increase the proliferation, migration, and invasion of RT4 cancer cells [52].

The role of CAFs in the EMT process and desmosomal remodeling has also been demonstrated in colorectal cancer [29,53]. The interaction of exosomes secreted by CAFs promotes cell invasion and chemotherapy resistance through the upregulation of exosomal miR-92a-3p [29].

According to Kangdi Li et al., breast cancer cells overexpress survivin (a protein that inhibits apoptosis) and secrete it into the extracellular environment via exosomes, which are then internalized by CAFs. CAFs increase the expression of SOD1 protein and transform into myofibroblasts, which in turn promote breast cancer cell proliferation, EMT, and the formation of cancer stem cells [54]. As observed in a colorectal cancer model, primary tumors release extracellular vesicles containing ITGBL1 (integrin beta-like 1) into the circulation, which activate fibroblasts residing in distant organs. Activated through the TNFAIP3/NF-κB signaling pathway, fibroblasts induce the formation of a premetastatic niche and promote tumor growth by secreting proinflammatory cytokines, such as IL-6 and IL-8 [55]. 

Obtaining technological control over exosomal intercellular information transfer may be a factor significantly limiting the protumor effect of CAFs.

### 3.4. Tumor-Associated Macrophages (TAMs)

For survival, a cancerous tumor needs protection, which involves, among other factors, polarized M2 macrophages.

The role of exosomes as an important mediator in the interaction between cancer cells and the microenvironment and a factor causing the polarization of macrophages toward the M2 phenotype stimulating immune suppression and tumor progression was confirmed by the research of Wang X. et al. [8]. It was shown that exosomes isolated from pancreatic cancer cells express miR-301a-3p under hypoxic conditions and can polarize macrophages via the PTEN/PI3Kγ signaling pathway. As a result, there is the migration of pancreatic cancer cells, invasion, and epithelial–mesenchymal transition in vitro, as well as the development of lung metastases, as confirmed by in vivo experiments conducted on male BALB/c nude mice. 

In a study conducted by Wei C. et al., it was noted that CRC cells derived from patients after the resection of a colorectal tumor undergo an EMT process mediated by tumor cells circulating in the blood (CTCs), and this entire process is regulated by TAMs. TAMs cultured with CRC cells regulated the EMT process by increasing the migration and invasion of CRC cells through the secretion of IL6, which was accompanied by the increased expression of vimentin and the decreased expression of E-cadherin. Moreover, it was shown that the stimulation of CRC cells with IL6 from TAMs increased the expression of the p-JAK2 and p-STAT3 signaling pathway, which activates the EMT process [56]. 

Research conducted by Lin F. et al. suggests that exosomes derived from a human bladder cancer cell line activate the program for macrophage polarization from the M0 to M2 phenotype, as evidenced by the production of specific cytokines, mainly IL-10 and TGF-β, and phenotypic changes in surface markers [57]. It was noticed that mi-RNA may play a special role in the communication of exosomes with TAMs. After the phagocytosis of exosomes derived from cancer cells containing miR-21 by macrophages, they were polarized toward an immunosuppressive phenotype (M2). In turn, cancer cells incubated with macrophages previously exposed to the content of exosomes showed a significantly greater ability to invade and migrate than cancer cells incubated with macrophages not subjected to this exposure. 

### 3.5. Myeloid-Derived Suppressor Cells (MDSCs)

MDSCs have long been described as one of the main factors inhibiting the function of effector cells in cancer [58]. MDSCs are immature myeloid cells that exhibit strong immunosuppressive activity, and two main categories have been identified: monocytic (M-MDSC) and polymorphic (PMN-MDSC). The differentiation of M-MDSCs into macrophages and dendritic cells is shaped by the tumor microenvironment. Many factors, such as hypoxia, the STAT3 transcription factor [59], growth factors, and other cytokines (e.g., IL-1β, IL-6, IFN-γ, IL-4, IL-13, and IL-10) [60,61] induce the differentiation of M-MDSCs into TAMs. Exosomes, secreted by both human and murine MSCs, accelerate the progression of breast cancer by inducing M-MDSCs into highly immunosuppressive macrophages (M2) within the tumor [62]. The immunosuppressive polarization of macrophages may also occur under the influence of exosomes secreted by cancer cells [57,63].

### 3.6. Mesenchymal Stem Cells (MSCs)

The polarization of procancer M2-like macrophages may occur in the presence of mesenchymal stem cells present in the tumor environment. A study was designed using cells derived from gastric cancer (GC-MSCs). After 3 days of the coculture of macrophages with GC-MSCs, macrophages showed higher levels of expression of the markers characteristic of the M2 phenotype. In addition, there was a significantly increased secretion of IL-6, IL-10, VEGF, and MCP-1, which promote tumor progression. IL-6 and IL-8 secreted by GC-MSCs were responsible for the polarization of macrophages through activation of the JAK2/STAT3 signaling pathway [64]. 

### 3.7. Cancer Stem Cells (CSCs)

Pleiotrophin (PTN) produced by TAMs is responsible for, among other effects, promoting cell proliferation, migration, and the induction of angiogenesis, and it contributes to increasing the percentage of cancer stem cells through the PTN/β-catenin pathway [65]. The communication between CSCs and macrophages is reciprocal: CSCs inhibit the proinflammatory properties of macrophages, which in turn initiate the establishment of CSC properties in tumor cells [65,66]. Another model of cellular interaction is based on the involvement of the GPNMB protein, which initiates the development of features characteristic of CSCs in cancer cells [67]. GPNMB is also secreted by tumor-associated macrophages (TAMs). In in vivo studies on a mouse model, cells producing GPNMB grew much faster and formed lung metastases, and the mRNA level of pluripotent genes was higher than in tumors without the GPNMB component [66]. 

### 3.8. Progression

In cancer processes, exosomes may participate in cancer growth and metastasis by regulating the immune response, blocking epithelial–mesenchymal transition, and promoting angiogenesis [68]. Exosomes can also aid in the creation of premetastatic niches, help tumor cells evade immune surveillance, and enable cancer cells to penetrate and colonize distant organs [69]. Research into the particular roles of exosomes in the development of cancer is progressing. Many studies have shown a close relationship between exosomes and the development and progression of cancer [70,71,72]. 

#### 3.8.1. Increase in Malignancy

One of the mechanisms for increasing the metastatic potential as a result of tumor progression involves pathogenic mtDNA (mitochondrial DNA) contained in exosomes. MtDNA originating from tumor cells with metastatic properties is transferred, via exosomes, to tumor cells characterized by lower metastasis and to stromal cells, both in vitro and in the tumor microenvironment, as observed in a mouse model [73]. 

There are also studies in which noncoding RNA (ncRNA) contained in exosomes was observed to promote the proliferation, invasion, angiogenesis, and migration of gastric cancers (GCs). Wei S et al. have shown that exosomal miR-15b-3p promotes the migration and invasion of GCs by targeting DYNLT1, caspase-3, and caspase-9 [74].

#### 3.8.2. EMT in Tumor Metastasis

The process of epithelial–mesenchymal transition driven by pleiotropic transcription factors, i.e., Snail1, Snail2/Slug, ZEB1, ZEB2, and Twist, is the main pathogenetic factor in the metastasis of cancer cells. The basis of this process is the elimination of intercellular cohesion, thus resulting in increased cellular mobility and the acquisition of migratory and invasive potential [75,76]. The tumor microenvironment plays an important role in the development of cancer metastasis by regulating the EMT, but it is still unclear by what mechanisms this process occurs [56,77]. 

In a study conducted by Jafari N et al., it was shown that the thrombospondin family protein TSP5, associated with cancer progression, found in exosomes isolated from the fat tissue of women with type 2 diabetes partially contributed to the EMT in breast cancer patients. As expected, TSP5-enriched exosomes had significantly increased transcription of *ZEB1* and *SNAI2* readout genes compared to the control [78]. 

### 3.9. Reprogramming of Metabolism

The mechanisms of tumor microenvironment communication involving CAFs remain unclear [79,80,81]. Metabolic pathways involving kinases may play a role in modulating the effect of CAFs on tumor promotion. An example is focal adhesion kinase (FAK), whose reduced expression level in tumor-associated fibroblasts is associated with changes in signaling and increased tumor growth. The mechanism of strengthening discrete metabolic pathways in CAFs may be important, thus leading to an increase in the production of chemokines that activate protein kinase A in cancer cells. The result is an increase in glycolysis and metabolic reprogramming toward cancer metabolism [82]. Another important element involved in reprograming metabolism is cancer-associated adipocytes (CAAs). Wang S. et al. found that exosomes isolated from in vitro mesenchymal stromal/stem cell (MSC) differentiated adipocytes promote breast cancer cell MCF7 proliferation and migration. Adipocyte exosomes also activated two key downstream effector proteins of Hippo (one of the major pathways controlling tumorigenesis), YAP, and TAZ. Furthermore, the depletion of exosomes in an in vivo mouse xenograft model attenuated the impact of adipocytes on tumor growth. Following the blocking of exosome MSC differentiated adipocytes, there was a decrease in the number of proliferating cancer cells, namely MCF7 (Ki67-positive cells) [83]. Another important component in the tumor microenvironment (TME) is cancer-associated endothelial cells (CAECs) [84]. It was demonstrated in a recent study that cancer cell exosomes could promote the process of glycolysis in endothelial cells. In their investigation, Wang B. et al. showed that exosomes secreted by acute myeloid leukemia cells promote proliferation, migration, and tube-forming activity in human umbilical vein endothelial cells (HUVECs). The contribution of glycolysis to the increased malignancy of cancer cells was confirmed on the basis of an increased value of glycoPER (a real-time indicator of changes in glycolysis rates) after treatment with exosomes from AML cell lines [85].

### 3.10. Impact on the Immune System

The immunological changes occurring in the bone marrow of mice caused by exosomes derived from multiple myeloma cancer cells were studied by Lopes R. et al. [86]. It was shown that exosomes derived from the MOPC315.BM multiple myeloma tumor line promote immunosuppression by modulating the phenotype of lymphoid cells, accompanied by an increase in the expression of PD-1 and CTLA-4 and a decrease in CD27, thus facilitating escape from the surveillance of the immune system and the progression of myeloma cells [86]. 

## 4. Exosome Technology: An Overview

Observation of the cancer process, leading to understanding the mechanisms underlying the creation of a protumor living space, is the key to developing effective therapeutic protocols. Table 2 presents a list of exemplary miRNAs and their associated activities that alter the malignancy of the cancer process, which can be utilized and have a potential effect in regulatory therapy. The subject of regulatory action is the expression of a protein/proteins with the potential to influence the biological malignancy of a tumor. A regulatory effect can be achieved by using a dedicated miRNA (Table 2: Method of action). All proteins presented in the table, via regulation of their expression, have the potential to modulate cancer malignancy (Table 2: CM). The authors of the referenced studies examined the impact of such action on such attributes of the cancer process as migration and invasion (Table 2: M/I), proliferation and apoptosis (Table 2: P/A), metastatic potential (Table 2: Meta), EMT process (Table 2: EMT), sensitivity to chemotherapy and radiotherapy (Table 2: CTH/RTH * sensitivity), and macrophage polarization to the M2 type (Table 2: M0/M2). Descriptions of regulatory effects from publications with a relatively high citation index were selected for this review.

The possibility of influencing the expression of signaling proteins on which the malignancy of a tumor depends seems to be a steadfast topic of scientific interest for researchers, and the intentional modification of the biological malignancy of a tumor is certainly not an exclusive experience (Table 2). Regulatory action, capable of reproducing the original conditions that are unfavorable for the existence of the cancer process, may prove to be an alternative to population treatment. We decided to check the current access to technology enabling regulatory intervention in the cancer process using exosomes.

### 4.1. EMT and Microenvironment

Attempts are being made to modify the expression of genes located in exosomes [100]. Research conducted by Wang X et al. involved silencing miR-301a-3p in exosomes isolated from pancreatic cancer using CRISPR/Cas9 [8,101]. In order to determine its role in the EMT process, Wang L. et al. conducted the knockdown of exosomal miR-19b-3p through the infection of ACHN and 786-O cancer cells with the miR-19b-3p lentivirus [40]. Attempts to regulate the expression of miR-6780b-5p and miR-92a-3p in the studies of Jing Cai et al. [38] and Yang B. et al. [90] involved the treatment of exosomes with agomir and antagomir, in the form of labeled and chemically modified microRNAs, which are improved miRNA mimics and inhibitors [102].

#### 4.1.1. Cancer-Associated Fibroblasts (CAFs)

Regarding squamous cell carcinoma cells of the tongue, exosomes containing miR-382-5p, isolated from a medium cultured with CAFs, had a stronger effect on the increase in the concentration of metalloproteinases MMP-3 and MMP-9, as well as N-cadherin and β-catenin, compared to exosomes secreted by normal fibroblasts. They also increased the migratory and invasive abilities of cancer cells and reduced their sensitivity to cisplatin [94].

#### 4.1.2. Tumor-Associated Macrophages (TAMs)

Jiang Z et al. experimentally inhibited the secretion of exosomes using GW4869 (a noncompetitive sphingomyelinase inhibitor), which is commonly used as an exosome inhibitor. Using a mouse model, they achieved a reduction in macrophage differentiation to an immunosuppressive phenotype and function (M2), with the effect of inhibiting tumor growth [63]. Both in a mouse model and in human cancer cells, miRNA influenced the AKT/STAT3 pathway [57,63].

#### 4.1.3. Bone Marrow Stem Cells (BMSCs)

The effect of exosomes from BMSCs on the promotion of tumor growth and metastasis was demonstrated both in vitro and in an in vivo mouse model [91,103]. Exosomes derived from BMSCs may contain noncoding RNAs (ncRNAs) of various types: long noncoding RNAs (lncRNAs), microRNAs (miRNAs), or circular RNAs (circRNAs). Their effect on cancer cells has not yet been fully characterized [35,82,97]. An attempt at such an assessment was conducted by Yuan Yao et al. Through in vitro studies conducted on a group of rabbits, it was shown that exosomes from BMSCs participate in the repair of damaged endometrial epithelial tissues by reversing the EMT process induced by TGF-β1. Endometrial cells treated with exosomes from BMSCs showed an increased expression of CK-19 and E-cadherin proteins with a reduced expression of VIM, FSP-1, TGF-β1, Smad 2, and P-Smad 2, and increased intercellular adhesion was observed, which supports the reversal of transformation from epithelial–mesenchymal to mesenchymal–epithelial transition (MET) [104].

### 4.2. Long Noncoding RNA (LncRNA)

Both exosomal lncRNAs and miRNAs are currently considered to be biomarkers of some malignancies, including in bladder and prostate cancer [34,105,106]. Exosomes produced by cancer may contain, among other factors, a noncoding RNA that initiates lymphangiogenesis and lymph node metastasis [107,108]. Following its internalization into exosomes, lncRNA epigenetically regulates gene expression (including SOX18/PROX1) by recruiting hnRNP (heterogeneous nuclear ribonucleoproteins) and increasing the level of H3K4 trimethylation in gene promoters. The packaging of long noncoding RNA into exosomes may occur through a mechanism called SUMOylation, among others [108]. Some authors point to the phenomenon of reducing the malignancy of cancer cells via the so-called mechanism of “sponging”. The accumulation of lncRNA or competitive miRNA, delivered in exosomes to cancer cells, inhibits their proliferative properties, migration, and invasion, as well as promotes their apoptosis [82]. 

### 4.3. Relationship with the Immune System

Cancer immunosuppression mediated by exosomes was confirmed in a study by Dou D et al. Exosomes derived from the CAFs of breast cancer patients promoted an increase in PD-L1 expression in cancer cells as a consequence of which they induced T cell apoptosis and impaired the function of NK cells. The effect of immune suppression induced by exosomes derived from CAFs was accompanied by an increased expression of miR-92 in breast cancer cells [109]. Chen G et al. studied the communication of exosomes with the immune environment and found that exosomes derived from the human melanoma cell line MEL624 were characterized by an increased expression of PD-L1, directly inhibiting the antitumor function of CD8 T cells (proliferation and cytotoxicity), as evidenced by a reduced expression of Ki-67 and granzyme B (GzmB) and an inhibited production of IFN-γ, IL-2, and TNF-α [110]. 

#### Tregs (T-Regulatory Cells)

Another example of the impact of exosomes on cells is a mechanism dependent on signaling from their surface, which does not require the internalization of exosomes. An example of such an exosomal influence would be T-regulatory cells (Tregs). Tregs are immunocompetent cells involved in tumor escape from immune surveillance through suppressing effector cells of the immune system. Jóźwicki et al. showed an increased presence of Tregs in the initial phase of development of urothelial cancer of the urinary bladder, which may be related to the facilitated development of immunological tolerance to the presence of the tumor. In the advanced stage of cancer, the number of Tregs was significantly reduced [111,112]. It has been shown that the suppressor functions of T-regulatory cells are modulated by exosomes derived from human cancer cells. Muller L. et al. showed that the signals produced by exosomes isolated from HNSCC (head and neck squamous cell carcinoma) modulate gene expression in human Tregs, and the highest transcriptional activity was observed in immunoregulatory genes such as PD-L1, PD-1, CD40L, CD25, and ZAP-70 [113].

### 4.4. Technology of Obtaining and Modifying Exosomes

Because exosomes are native to an organism, their surface has biochemical properties similar to those of the organism’s cells. This allows them to avoid phagocytosis and lysosomal uptake, instead fusing with cell membranes [114], and this is a basis for why these exosomes of native origin elicit a low immune response [115]. It is possible to obtain exosomes directly from dedicated cells, thus defining their specific biological behavior and action. Another possibility is by extrusion to induce the formation of exosome-like vesicles made of cell membranes. Depending on the method of acquisition, exosomes can be biologically modified in terms of surface and content, thus resulting in their specific functionality (Figure 1).

However, the use of exosomes as potential therapeutic carriers is associated with numerous difficulties. The main technical obstacles are their small size and surface complexity, which necessitate the use of complex processing processes that may reduce their biocompatibility compared with currently available methods [116]. Some substances can be incorporated into exosomes by direct incubation. Using this method, Pablo Lara et al. determined the distribution, selectivity, and effectiveness of lipophilic zinc phthalocyanine PS (ZnPc), a photosensitizing substance that was placed in exosomes of native origin and then added to a coculture model simulating the tumor microenvironment. Preferential uptake of exosomal ZnPc by colorectal cancer cells over macrophages and dendritic cells was noted [117]. This experiment enabled the inclusion of exosomes in the group of carriers of signaling molecules and the search for methods to intentionally modify their content (Figure 2). 

Table 3 presents the available technologies for obtaining and modifying the content of exosomes and exosome-like nanovesicles and the attempts to use them as documented in the literature.

#### 4.4.1. Exosomes Loaded with Chemotherapy Drugs

The use of exosomes as a cargo vector for chemotherapeutic drugs resulted in the improved effectiveness of chemotherapy, which was probably due to specific interactions in recipient cells and optimized endocytosis mechanisms [133]. In an animal model of breast cancer, Hadla Mohamet et al. showed that exosomal doxorubicin (DOX) reduced cardiotoxicity due to the limitation of DOX penetration through endothelial cells into the myocardium, which allowed for the administration of a higher dose of the drug [134]. Aquil Farrukh et al. conducted a study in which celastrol (CEL) was transported with the help of exosomes. It was observed that compared to free CEL, the CEL administered in this way was more effective against non-small cell lung cancer, thus inhibiting the proliferation of A549 and H1299 cell lines [135].

#### 4.4.2. Exosomes for Delivery of Functional Proteins in Cancer Therapy

The unique properties of exosomes also enable proteins to be transported while maintaining their stability and bioactivity. Protein/peptide drugs or target signaling proteins have superior pharmacokinetic properties when encapsulated in an exosomal carrier thanks to the ability to penetrate target tissues, which may significantly contribute to increasing, e.g., the drug bioavailability [136]. Exosomes seem to be a promising vehicle for delivering biomolecules to recipient tissues (Table 4), but the mechanisms enabling their exosomal packaging are still not well understood.

## 5. Discussion 

Knowledge allows you to plan an experiment and observe the effects, while technology allows you to plan and achieve the effect. The path toward regulatory therapy in oncology will certainly not be easy. The transition from knowledge to technology takes time. Are we ready for this? In practice, we see relatively simple attempts at the limited use of technologies aimed at purposefully modifying the malignancy of the cancer process. However, the analysis of the techniques used allows for some moderate optimism. A possible starting point is the observation that the biological malignancy of a tumor may be associated with the expression of a gene or protein. By analyzing the available data in the HPA database, we were able to indicate the cut-off value of expression of a selected gene in a given tumor (FPKM), thus allowing for the stratification of patients into two groups with significantly different 5-year survival times (Table 1). It cannot be ruled out that gene/protein expression should be the primary goal of technological action. The added value of this observation is that the obtained FPKM threshold value may be used as an orienting factor in the optimization of gene/protein expression modulation technology. However, the technology used to influence gene/protein expression is dependent on a signal carrier. A good candidate seems to be the exosome, which was first described in the 1980s in sheep and rats. Since then, understanding of the biological importance of exosomes has changed significantly. It can be said that their perception has changed dramatically from being considered junk [142,143,144] to crucial in intercellular communication [145,146,147,148,149,150,151]. The most important stages of cancer development—initiation, promotion, progression, metabolic reprogramming, and escape from immune surveillance—are based on intercellular epigenetic relationships. As a key to the life of the tumor, they have developed as a result of the dialogue between tumor cells and microenvironmental cells, such as cancer-associated fibroblasts (CAFs) or tumor-associated macrophages (TAMs) [152,153,154]. From the available literature, it is known that exosomes are present in the pathomechanism of all these stages. Most likely for this reason, the interest of researchers has recently been focused on attempts to influence the malignancy of the cancer process by modulating gene/protein expression using exosomes as a carrier of signaling particles. Table 2 presents examples of this research, which have been selected based on the high degree of interest as assessed by the citation index of the selected publications. The technological effect on the malignancy of various cancers was assessed by comparing similar parameters characterizing the cancer process, such as migration, invasion, proliferation, apoptosis, metastatic potential, EMT process, sensitivity to chemotherapy and radiotherapy, and macrophage polarization to the M2 type. The analysis of the presented data allows us to notice a relatively constant impact of technological activity in terms of the EMT parameter (Table 2: EMT), the inhibition of which was associated with a reduction in tumor malignancy (Table 2: CM) in most studies. In terms of other parameters, no similar relationships were apparent. It cannot be ruled out that the process of epithelial–mesenchymal transformation of a cancer cell may be a good first candidate for future regulatory therapy. It is worth noting that each change in the expression of a dedicated miRNA was associated with an effect on cancer malignancy, regardless of the molecularly defined pathomechanism of the effect involving a single protein, protein family, or signaling pathways (Table 2: Action). For this reason, the assessment of the effectiveness of the technology used in these studies is not sufficiently informative. It is unclear how the effectiveness of regulatory technology may be affected by the number of proteins whose expression is regulated. The consideration of proteins involves not only rewriting the code but also accounting for posttranscriptional and posttranslational modifications, which are subject to various epigenetic modifiers. Many studies focus on attempts to influence cancer malignancy by regulating the expression of a single protein, which may not be sufficient to achieve an effectively stable regulatory therapy. However, this situation is changing rapidly. Researchers are interested in technologies that enable modification of not only the EMT process but also the dialogue between cancer cells and elements of the tumor microenvironment (CAFs and TAMs), technologies based on noncoding RNA, and those for influencing the relationship with the immune system. The focus of this interest indicates potential additional targets for regulatory impact. An element of improving used and future regulatory technologies is the development of methods for obtaining and modifying exosomes (Table 3). Large amounts of exosomes with an increased ability to transport and provide signaling molecules to target cells will most likely constitute a basic element of regulatory technologies in the near future (Figure 1 and Figure 2). For practical use, solutions are also available to replace exosomes with other nanovesicles and combine them into larger complexes with greater effectiveness (Table 4). Although the available technological possibilities for regulatory therapy, as well as their use, are still in their infancy, it seems that we have started marching in the right direction. There are still no projects in which the problem of regulatory influence on cancer cells would be treated in a more multifaceted manner, keeping in mind that a cancer cell has just escaped the control of the entire well-functioning immune system. Moreover, if cancer care (promotion) mechanisms are present in the host, attempts to inhibit the signaling formula of a single protein will most likely not result in successful regulation. We know that it is necessary to start identifying targets for cancer therapies in the category of protein regulatory networks [155,156,157]. Attempts to simultaneously regulate the expression of proteins that have a multipoint influence on promotion (EMT process, acquisition of the stem cell phenotype, mobilization of CAFs and TAMs, etc.) and progression (metastasis) may prove to be a valuable link in the effectiveness of regulatory therapies.

## 6. Current Challenges and Future Outlook

Despite the tremendous therapeutic and diagnostic promise, multiple hurdles are to be faced in realizing the potential of exosomes. Examples of directions to address technical issues include improving isolation methods, achieving scale-up, and ensuring reproducible engineering. Cargo space and rapid clearance limit therapeutic applications. Combining exosomes with more potent immune adjuvants could boost outcomes. Regulatory challenges include establishing appropriate safety, potency, and manufacturing guidelines. Addressing these challenges will accelerate the advancement of engineered exosomes from preclinical studies to cancer patients.

## 7. Conclusions

In summary, tumor-derived exosomes are important intercellular communicators that contribute to many facets of cancer progression. However, engineering exosomes to counteract their endogenous tumor-promoting functions represents a promising new frontier in developing naturally targeted and biocompatible nanotherapeutics for cancer. Overall, the ability to engineer and modify tumor-cell-derived exosomes offers unique opportunities to study cancer biology, as well as develop more targeted diagnostic and therapeutic strategies against cancer. Further research should focus on achieving upscaled production and reproducible modification of exosomes, elucidating cargo loading and delivery mechanisms, and evaluating therapeutic efficacy and safety through in vivo studies. Advancing extracellular vesicle engineering could enable the development of “intelligent” drug delivery systems that take advantage of the innate biology for more precise and potent anticancer therapies. 

## Figures and Tables

**Figure 1 cancers-16-00923-f001:**
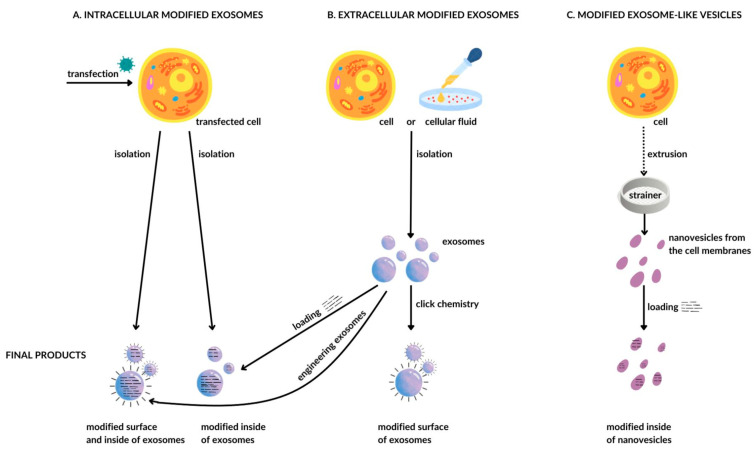
Classification of exosome modification methods: (**A**) intracellular before extraction, (**B**) extracellular after extraction from the cell, and (**C**) modification of exosome-like vesicles obtained by extrusion from cell membranes.

**Figure 2 cancers-16-00923-f002:**
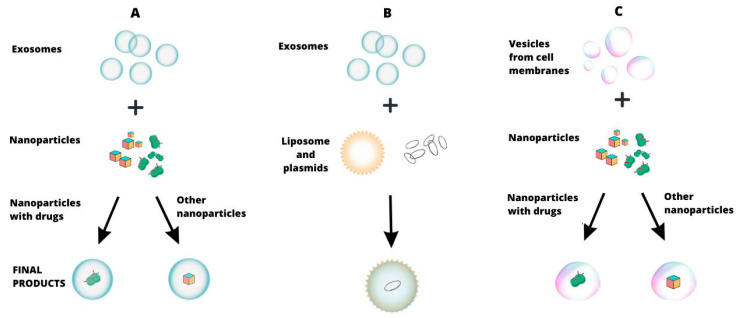
Content modulation of exosomes and exosome-like vesicles (obtained by extrusion from cell membranes) by direct incubation: (**A**,**C**) nanoparticles and drugs; (**B**) liposomes and plasmids.

**Table 1 cancers-16-00923-t001:** Statistically significant correlation of the expression of selected genes with 5-year survival in cancers from the Human Protein Atlas database.

Potential Therapeutic Use	Coding Gene//Protein/Publication	Average/ Maximum FPKM Value	Best Expression Cut Off (FPKM)	5-Year Survival High/Low [%]	*p*-Score
Colorectal cancer	CREB1//CAMP responsive element binding protein 1/ [27]	3.9/27.0	3.2	66/49	0.033
	ZEB1//Zinc finger E-box binding homeobox 1/ [28]	1.9/26.8	2.35	57/63	0.034
	FBXW7//F-box/ [29]	1.7/9.3	1.71	68/56	0.018
Breast cancer	VEGFB//Vascular endothelial growth factor B/ [30]	48.5/296.1	44.57	85/79	0.049
	SMARCA5//SWI/SNF related/ [31]	17.7/55.4	21.43	77/83	0.0092
	SNAI2//Snail family transcriptional repressor 2/ [32]	10.2/183.1	6.55	85/77	0.035
Urinary bladder carcinoma	SNAIL1//Snail family transcriptional repressor 1/ [33]	3.1/57.1	0.77	38/56	0.019
	IGF2R//Insulin-like growth factor 2 receptor/ [34]	9.1/27.9	8.89	30/50	0.00061
	ABL2//ABL proto-oncogene 2/ [35]	2,5/13,5	2.03	33/50	0.00084
Ovarian carcinoma	SNAIL1//Snail family transcriptional repressor 1/ [36,37]	2.9/21.2	2.8	20/40	0.0098
	Notch-1//Notch receptor 1/ [38]	7.1/ 61.1	10.2	25/34	0.0055
Gastric cancer	ZEB1//Zinc finger E-box binding homeobox 1/ [39]	6.4/51.8	6.15	13/45	0.0056
	ZEB2//Zinc finger E-box binding homeobox 2/ [39]	2.1/9.2	1.76	24/48	0.011
Renal cell carcinoma	PTEN//Phosphatase and tensin homolog / [40]	8.7/37.8	6.05	66/77	0.00073
Hepatic cell carcinoma	Twist1//Twist family bHLH transcription factor 1/ [41]	0.4/45.1	0.14	37/52	0.018
	FOXQ1//Forkhead box Q1/ [42]	4.2/79.4	2.6	39/52	0.033
Uveal melanoma	CDK4//Cyclin dependent kinase 4/ [43]	38.9/176.2	34.4	0/57 *	0.0032

* 3-year survival high/low [%].

**Table 2 cancers-16-00923-t002:** Modification of miRNA action in cancer cells and potential therapeutic benefit.

Method of Action		Potential Therapeutic Benefit									Y/CI *
miRNA	Action *	Effect of Modification of miRNA Expression								Clinical Application	
		miRNA	Inhibition				CTH/RTH * Sensitivity	M0/M2 *	CM *		
			M/I *	P/A *	Meta *	EMT					
miR-134	IR: CREB1	+	+/+	nd/nd	nd	+	+	nd	-	CRC [27]	2017/52
miR-100	mTOR/HIF-1α/VEGF modulation	+	+/+	nd/nd	nd	+	nd	nd	-	BC [30]	2017/194
	IR: HOXA1 and SMARCA5	+	+/+	nd/nd	nd	-	nd	nd	-	BC [31]	2014/101
miRNA-34b/c	IR: β-katenin	+	+/+	nd/nd	nd	+	nd	nd	-	PC [87]	2015/116
miR-30c	IR: SNAIL1	+	+/+	nd/nd	nd	+	nd	nd	-	BC [33]	2018/221
miR-137	IR: Snail	+	+/+	nd/nd	nd	+	nd	nd	-	OC [36]	2016/89
MiR-363	IR: Snail	+	nd/nd	nd/nd	nd	+	+	nd	-	OC [37]	2018/29
miR-452	IR: SNAI2	+	nd/+	nd/nd	nd	+	nd	nd	-	BC [32]	2017/58
miR-200c	IR: Zeb1, Zeb2	+	+/+	nd/nd	nd	+	+	nd	-	GC [39]	2018/56
miR-361-5p	IR: Twist1	+	+/+	+/nd	nd	+	nd	nd	-	HCC [41]	2020/9
miR-124	IR: PRRX1	+	nd/nd	nd/nd	nd	+	+	nd	-	CRC [88]	2014/93
miR-155-5p	IR: GATA3	-	nd/nd	nd/nd	nd	+	nd	nd	+	GC [89]	2019/86
miR-4319	IR: FOXQ1	+	nd/nd	+/+	nd	+	nd	nd	-	HCC [42]	2019/19
miR-6780b-5p	RE: Notch/MAPK pathway	-	+/+	+/nd	nd	+	nd	nd	+	OC [38]	2021/20
miR-19b-3p	RE: PTEN IR: E-cadherin, IN: N-cadherin, vimentin, Twist protein, CD103	-	nd/nd	nd/nd	+	+	nd	nd	+	RCC [40]	2019/98
miR92a-3p	ID: Akt/Snail pathway	-	nd/nd	nd/nd	+	+	+	nd	+	HCC [90]	2020/79
miR-301a-3p	IR: PTEN, IN: PI3Kγ, p-AKT, p-mTOR	-	+/+	nd/nd	+	+	nd	nd	+	PAC [8]	2020/9
	IN: PTEN/ PI3Kγ pathway, Arg1, TGFβ, IL10	-	+/+	nd/nd	nd	nd	nd	+	+	PAC [8]	2020/9
miR-1500, miR-210-3p, miR-193	ID: STAT3, IR: E-cadherin, IN: snail, vimentin, slug, twist, fibronectin, ZEB1, N-cadherin	-	nd/+	nd/nd	nd	+	nd	nd	+	LC [91]	2019/246
miR-34a	IR: Snail	+	nd/+	nd/nd	nd	+	nd	nd	-	OC [36]	2016/89
	IR: ZNF281	+	+/+	+/nd	+	+	nd	nd	-	CRC [92]	2013/123
										HCC [93]	2013/1067
miR-375-3p	IN: E-cadherin, IR: Snail, vimentin, ZEB-1, β-catenin	+	+/+	nd/nd	nd	nd	nd	nd	-	CRC [28]	2021/22
miR-382-5p	IN: MMP-3, MMP-9, N-cadherin, β-catenin	-	+/+	nd/nd	nd	+	nd	nd	+	OSCC [94]	2019/81
miR92a-3p	IN: Wnt/β-catenin pathway, IR: MOAP1, FBXW7	-	+/+	nd/-	nd	nd	+	nd	+	CRC [29]	2019/299
miR-21-5p	IN: PTEN, STAT3 pathway, IL-10, TGF-β	-	nd/nd	nd/nd	nd	nd	nd	+	+	BDC [57]	2020/51
miR-155-5p	IR: ZC3H12B, IN: IL-6	-	nd/nd	+/-	nd	nd	nd	nd	+	CRC [95]	2021/29
miR-361-3p	IR: IGF2R	+	+/+	+/nd	nd	nd	nd	nd	-	BDC [34]	2021/9
miR-1224-5p	IR: CREB1	+	nd/nd	+/nd	nd	nd	nd	nd	-	BDC [96]	2020/23
miR-34b/c	IR: c-Met, (G1 cell cycle)	+	+/nd	+/nd	nd	nd	nd	+	-	Uveal MM [43]	2012/58
miR-205	IR: RHPN2	+	+/+	+/-	nd	nd	nd	nd	-	PC [97]	2019/45
miR-19b-1-5p	IR: ABL2, Bcl-2, MMP2, MMP9, IN: Bax	+	+/+	+/-	nd	nd	nd	nd	-	BDC [35]	2021/7
miR-382-5p	IR: PTEN, ATRA,	-	nd/nd	nd/nd	nd	nd	+	nd	+	APML [98]	2019/23
	RE: cyclin D1										
	RE: RERG	-	+/+	nd/nd	nd	nd	nd	nd	+	BC [99]	2017/55

* Legend. I. Abbreviations of titles and entries— APML: acute promyelocytic leukemia; BC: breast cancer; BDC: bladder cancer; CM: cancer malignancy; CTH/RTH: chemotherapy/radiotherapy; CRC: colorectal cancer; GC: gastric cancer; HCC: hepatocellular carcinoma; ID: induction; IN: increase; IR: inhibitor; LC: lung cancer; M0/M2: polarization of macrophages to M2 phenotype; Meta: metastases; M/I: migration/invasion; OC: ovarian cancer; OSCC: oral squamous cell carcinoma; PAC: panreatic cancer; P/A: proliferation/apoptosis; PC: prostate cancer; RCC: renal cell carcinoma; RE: regulation; Y/CI: year of publication/citing index. II. -: decreasing action; +: increasing action; nd: no data.

**Table 3 cancers-16-00923-t003:** Methods of modifying exosomes: loading and modifications of the exosomal membrane.

Type of Action	Method	Description of the Method	Weaknesses *	Strengths *	Example of a Packaged Substance
Extracellular exosome loading	Incubation with target cargo	Temperature 37 °C for 1 hour with shaking	LE	SP and MEMI	Paclitaxel [118]
	Sonication	UE–DEMI–LTC	ED, EA, and EF	HE and PCP	Paclitaxel [118] Catalase [119]
	Electroporation	ElF–DEMI–LTC	DEI, EA, EF, and ITL	SP, CP, and HE	Doxorubicin [120] Catalase [119]
	Extrusion	Exosomes and target cargo—FFDPS	ED	HE and PCP	Porphyrins [121]
	Freeze–thaw	METC—3 cycles of fast freezing and thawing	EA	HE and MF	Thymoquinone [122]
		Exosomes with liposomes—several freeze–thaw cycles			Membrane fusion [123]
	Modified CCM	METC—thermal shock	DNA and DEMI	SP and HE	miRNA [28]
	pH gradient-based method	Generation of pHG and incubation with target cargo	Unpredictable effect associated with DEMI	HE, possibility of reusing the cargo	miRNA packaging [124]
	Saponin	Increasing exosome lipid membrane permeabilization	Hemolytic activity	LHME	Porphyrins [121]
	Transfection	LmiRNAGV with exosomes—enhancement of miRNA expression	Limited rate of penetration through membranes	MGME and increasing the therapeutic effect	pre-microRNA [125]
	Transfection with lipofectamine	Increasing the efficiency of transfection of RNA or plasmid DNA into cell cultures in vitro	There is no known mechanism of action of lipofectamine	HE, “gold standard”, and low toxicity to cells	DNA [126]
Intracellular loading during exosomes biogenesis	Bioengineering of exosome-producing cells	Cells with target cargo	TCM, DSM, and unwanted cellular content in exosomes	HE, preservation of native features of exosomes, and low toxicity to cells	Taxol [127]
		Transfection into cells with miRNA/siRNA/pDNA/plasmid vector to increase gene product expression in the exosome			IL3-Lamp2b plasmid vector [128] and encoding the fusion protein [129]
Surface-modified exosomes	Click chemistry	Attachment of molecules to the surface of exosomes through covalent bonds	Few scientific reports	HE, CP, no effect on size, adhesion, and internalization of exosomes	Copper-catalyzed azide–alkyne cycloaddition [130]
	Combinations with pH-sensitive fusion peptides	Exosomes + fusion peptide = formation of pores in the lipid membrane due to lower pH	Complexity of the method, few scientific reports	Control of exosome movement, better presentation of tumor antigens	Exosomes from melanoma cells and mixing with GALA [129]
	Dual ligand engineering	Sonication and incubation of a mixture of exosomes, vector lipid molecules and target cargo	Identification of a vector molecule unique to a specific cancer	High ability to accumulate in cancer cells, the therapeutic effect	PTX-loaded exosomes with PEG-AA vector moiety [131]
Exosome-mimetic	Nanovesicles from cell membranes	The result of cell disintegration using extrusion	Efficacy depends on the surface properties of the cells used	Features similar to exosomes but 100 times higher production efficiency	Nanovesicles from monocytes or macrophages with doxorubicin [132]

* Legend: LE—low efficiency; HE—high efficiency; SP—simplicity of performing; ED—exosome deformation, EA—exosome aggregation; EF—exosome fusion; PCP—rotein cargo protection against degradation by proteases; ITL—instability of therapeutic loads; MEMI—maintenance of exosome membrane integrity; DEMI—disruption of exosome membrane integrity; CP—controllability of parameters; MF—membrane fusion between exosomes and liposomes; UE—ultrasonic energy; LTC—loading target cargo; ElF—electric field; FFDPS—filtering on filters with decreasing pore sizes; CCM—calcium chloride method; METC—mixture of exosomes and target cargo; DNA—destabilization of nucleic acids; pHG—pH gradient between the intra- and extraexosomal environment; LHME—loading of hydrophilic molecules into exosomes; LmiRNAGV—lentivirus pre-miRNA gene vector; MGME—modification of the genetic material of exosomes; TCM—time-consuming method; DSM—difficult to scale method.

**Table 4 cancers-16-00923-t004:** Exosome and exosome mimetics as a component of larger complexes.

Method	Particle Covered	Description of the Method	Weaknesses	Strengths	Application Examples
Covering a molecule with an exosomal membrane	Hybrid nanoparticles	Nanoparticles synthesized from inorganic and organic building block units, loaded with cargo, and coated with exosomes	The complexity of the method, few scientific reports	Exosomal transfer selectivity, preferential capture by specific cell types	Iron-based metal−organic framework nanoparticles with calcein or suberohydroxamic acid [137]
	Cationic nanoparticles	Incubation of exosomes with a mixture of synthetic cationic nanoparticles using the interaction of their electric charges	High costs, little scientific reports	Maintaining native EV features, comparable loading efficiency, lower toxicity	Nanoparticles with Cas9 protein [138]
	Drug-loaded silica nanocarriers	Combination of exosomes, drugs, and porous silica nanoparticles using acoustofluidics	Complexity of the method, few scientific reports	Drug loading and encapsulation in minutes	Silica nanocarriers with doxorubicin [139]
Covering a molecule with a cell membrane	Polymer nanoparticles	Combining membrane vesicles with PLGA (poly(lactic-co-glycolic acid)) particles using sonication	Complexity of the method, few scientific reports	Increased blood half-life and prolonged retention, decreased uptake by macrophages	Docetaxel-loaded PLGA nanoparticle [140]
Hybrids	Exosome–liposome	Incubation with the effect of fusion of exosomes with liposomes and encapsulation of plasmids	Few scientific reports	Effective encapsulation of large plasmids and drugs, reducing drug resistance	CRISPR/Cas9 vectors in an exosome–liposome hybrid [141]

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
