# Peer review of "Exosomes—Promising Carriers for Regulatory Therapy in Oncology"

_cancers, 2024, doi:10.3390/cancers16050923_

Round 1

Reviewer 1 Report

Comments and Suggestions for Authors

Review of the manuscript “Exosomes - Promising Carriers in Regulatory Therapy in Oncology” for the MDPI-Journal Cancers.

The authors present a review of molecular therapy studies for various forms of cancer. What these cited studies have in common is the targeting of exosomal signaling pathways in different cell types of the tumor and the tumor environment. On the positive side, it should be emphasized that this is a valuable overview of existing studies. However, the manuscript in its current form does not yet meet the expectations raised by the headline. Nevertheless, it seems possible to generate an article worthy of publication from this manuscript after extensive revision.

Therefore, acceptance of the manuscript for publication is recommended, provided that the following major revisions of the presentation style and content have been carried out.

1.       After the introduction, a detailed section should be provided on the current knowledge concerning exosomes in general, on all their variations and on their biological significance. For instance, microvesicles are mentioned in the abstract, but not in the text.

2.       The human protein atlas (HPA) is just mentioned one time on page 2. More information should be provided including the exact definition of the parameter FPKM.

3.       Numbering of section 2.4. (page 5 ff.) is misleading, because it summarizes different cell types under the heading “2.4 Tumor Associated Macrophages (TAMs)”. Please renumber afterwards in the following way: “2.5 Myeloid-Derived Suppressor Cells (MDSC)”, “2.6 Mesenchymal Stem Cells (MSCs)”, “2.7. Cancer Stem Cells (CSCs)”, “2.8 Progression”, “2.9 Reprogramming of the metabolism”, “2.10 Impact on the immune system”.

4.       Typesetting and information provided by table 2 (page 7-12) is unclear. What is the main information? The meaning of some abbreviations such as “nd” are missing in the legend on page 12, the information especially in column 2 from the left must be explained. In line 2 of the table, the short line breaks within single words look very inconvenient. In the present form, this table cannot be published. In contrast, table 3 on page 14-16 is good and should serve as a model for table 2.

5.       Many studies cited by the authors are not explained in a sufficient way. A good example are lines 375-378 about the T-regulatory cells. Here, only the abbreviation “Tregs” appears, the cell type is not explained and the results and conclusions of the study cited here remain unclear. More information should be provided in order to improve the clinical significance of the present review manuscript.

6.       A Discussion section is missing. Instead of the short section “4. Current Challenges and Future Outlook” on page 17, the authors fail to discuss the studies and their results provided in the previous sections in a detailed form. Which studies should be highlighted, which methods should be preferred, which cells types seem to be of major importance for targeted therapies, what about the differences between exosomes and microvesicles regarding their biological significance, and so on. All these aspects must be discussed. Without such a discussion section, the manuscript is just a sequence of different studies cited in the text and in the tables.

7.       Citation of the references should be re-checked carefully. For instance, in reference 1 the year is provided but not the page-numbers. In reference 6, just the volume and the year is mentioned. The correct citation style according to the instructions of the authors should be studied again and applied to all references. In the present form, it is unlikely that the bibliography will be accepted.

Reviewer 2 Report

Comments and Suggestions for Authors

Jóźwicka et al have done a literature survey on exosomes as emerging drug carries for applications in oncology. This is a timely topic and could be of interest to the readers of this journal. There are a few issues that need to be addressed before suggesting for publication.

The overall English needs an improvement.

The simple summary does not properly lay the reasonings as why exosomes and what is covered in this review paper. I suggest rewriting the abstract as the current one is ambiguous.

The following statement in the abstract “Extracellular vesicles (EVs), including exosomes and microvesicles, are emerging as important mediators of intercellular communication in the tumor microenvironment” is not quite accurate as communication role EV is not only limited to tumor microenvironment.

The introduction of this review is very short and would benefit from including other application of Exosomes such as biomarker for early diagnosis e.g. doi.org/10.1002/anbr.202300055.

In the section 4, could the authors elaborate more on the current challenges towards the application of Exosomes? This section is very brief and rather inconclusive.

The outlook and conclusion could have been expanded to further emphasize on the potential applications and translation of EV therapy in future.

Overall, each section of this review paper is a short summary of the different topics in the field. Although, form a review paper, it is expected a comprehensive survey of the current state of the art is provided rather than including a few examples.

Comments on the Quality of English Language

Moderate editing of English language required

Reviewer 3 Report

Comments and Suggestions for Authors

The review presented is interesting. However, there are some comments raised for your consideration:

1. Section 2.5~2.7 could be more thorough. For example, in Section 2.5.1 'Increase in malignancy', research has discussed the roles of exosomes in cancer malignancy in many aspects, including but not limited to promoting cancer proliferation, establishing a premetastatic niche, and regulating drug resistance. The author should integrate additional information and research findings to provide a more comprehensive overview in this section.

2. In the 'Current Challenges and Future Outlook' section, the authors should further discuss the specific challenges associated with the applications discussed in the paper, such as the use of miRNAs and exosomes as therapeutic carriers. A detailed examination of their limitations in treating specific types of cancer would be informative. Additionally, it would be valuable for the authors to summarize or review recent advancements and potential improvements made in these areas.

3. Table 2 needs to be revised. The current spacing makes it hard to read. Consider reorganizing or modifying column widths, adjusting the font size, or reorganizing the information to avoid overcrowding.
